# The epidemiology of antidepressant use in South Korea: Does short-term antidepressant use affect the relapse and recurrence of depressive episodes?

Min Ji Kim[1], Namwoo Kim[1], Daun Shin[1], Sang Jin Rhee[1], C. Hyung Keun Park [1,2,3], Hyeyoung Kim[4], Sung Joon Cho[5], Jae Won Lee[6], Eun Young Kim[7,8], Boram Yang [9], Yong Min Ahn [1,2]*

**1** Department of Psychiatry, Seoul National University Hospital, Jongno-gu, Seoul, Republic of Korea, **2** Department of Psychiatry and Behavioral Science, Institute of Human Behavioral Medicine, Seoul National University College of Medicine, Jongno-gu, Seoul, Republic of Korea, **3** Department of Psychiatry, Asan Medical Center, Seoul, Republic of Korea, **4** Department of Psychiatry, Inha University Hospital, Inhang-ro, Jung-gu, Incheon, Republic of Korea, **5** Department of Psychiatry, Kangbuk Samsung Hospital, Sungkyunkwan University School of Medicine, Seoul, Republic of Korea, **6** Dept. of Psychiatry, Seoul Metropolitan Eunpyeong Hospital, dept. of Psychiatry, Eunpyeong-gu, Seoul, Republic of Korea, **7** Mental Health Center, Seoul National University Health Care Center, Gwanak-gu, Seoul, Republic of Korea, **8** Department of Medicine, Seoul National University College of Medicine, Jongno-gu, Seoul, Republic of Korea, **9** Medical Research Collaborating Center, Seoul National University Hospital, Seoul, South-Korea

* aym@snu.ac.kr

**Data Availability Statement:** All relevant data are within the manuscript and its Supporting Information files.

## Abstract

### Background

The duration of antidepressant use affects the treatment of depression. Using the National Health Insurance database, which covers almost the entire national population, we verified the factors associated with the inadequate short-term use of initially prescribed antidepressants and their effects on the relapse and recurrence of depressive episodes.

### Methods

There were 752,190 patients included who had been newly prescribed antidepressants in 2012 with the diagnosis of depressive disorder. They were followed-up until December 31, 2015. They were classified as short-term and long-term antidepressant users depending on whether they used a specific initial antidepressant for at least four weeks. Sociodemographic, clinical, and medical utilization factors affecting the duration of antidepressant use were investigated. We also identified whether the duration of antidepressant use affected the risk of relapse and recurrence, which was defined by the restarting of antidepressants.

### Results

Initial antidepressants were taken for less than 28 days by 458,057 (60.84%) patients. Tricyclic antidepressants were used as the initial antidepressant more frequently than selective serotonin reuptake inhibitors (64.5% versus 19.3%). The type of initial antidepressant,

**Funding:** The authors received no specific funding for this work.

**Competing interests:** The authors have declared that no competing interests exist.

polypharmacy, psychiatric and medical comorbidities, type of insurance coverage, and type of medical institution visited were associated with short-term use. Short-term use marginally increased the risk of relapse and recurrence of depressive episodes (Hazard ratio: 1.06, 95% confidence intervals 1.048–1.075).

## Conclusions

Short-term antidepressant use is widespread in Korea, and assessment in various aspects are necessary to set proper treatment plans.

## Introduction

Depressive disorders are among the most common and burdensome psychiatric disorders worldwide causing severe disabilities in the occupational and social life of patients. Due to their chronic course and highly recurrent nature, they have significant personal and public health consequences[1]. According to the World Health Organization, depression is predicted to become the second most burdensome disease in 2020 [2]. It is obvious that long-term strategies should be made to decrease the burden of major depressive disorders considering relapses and recurrences. However, the inadequate antidepressant treatment and frequent relapsing nature of these disorders remain significant obstacles [3].

The treatment guidelines for major depressive disorders published by the American Psychiatric Association recommend a three-phase antidepressant treatment for depressive episodes. The acute phase treatment should last for at least 6–8 weeks, and should be followed by a continuation phase lasting for 4–9 months beyond the acute phase to consolidate the treatment response and reduce the risk of relapse [4, 5]. However, there is growing evidence that the treatment guidelines are often not followed. Approximately 30–40% of patients discontinue antidepressant treatment early in the acute phase of their treatment, even within the first few weeks of its initiation [6–9]. A previous study among Korean participants confirmed that 43.5% of patients at university hospitals stop medications within six weeks [10].

Little is known about the factors affecting the early termination of antidepressants. Previous reports show mixed results on whether sociodemographic characteristics tend to predict the risk of discontinuing antidepressants early [7, 11, 12]. There is some evidence that the initial choice of medication influences the duration of antidepressant treatment, depending on the tolerability of the drug. Patients prescribed a tetracyclic antidepressant (TCA) tend to discontinue their treatment earlier than patients prescribed a drug from a newer class of antidepressants [13–16]. However, further understanding of the determinants of the duration of antidepressant treatment in more generalized populations is needed.

There were several previous studies that use administrative data to assess whether the early discontinuation of antidepressants influences the treatment outcome by sorting patients based on the operational criteria of the number of prescriptions or the durations of treatment, ranging from 75 to 90 days.[17–20]. They consistently show that patients who discontinued their antidepressants during the continuation phase were more likely to experience a relapse or recurrence than patients who continued using their drugs. However, studies using administrative data rarely focused on patients who were not able to continue antidepressants for at least four weeks and exclude them from the study population [19].

The aim of this study was to describe the current status of antidepressant use in South Korea, especially the initial prescription, and to find the factors associated with short-term antidepressant use in the entire Korean population. We simultaneously explored whether

short-term antidepressant use influences the relapses and recurrences of the depressive episodes in the future.

## Materials and methods

### Subjects

Approximately 98% of South Korean people are registered in a universal health coverage system of National Health Insurance (NHI)[21]. The Health Insurance Review and Assessment Service (HIRA) provides the claims database containing information on diagnoses of patients coded using the International Statistical Classification of Disease and Related Health Problems 10[th] revision (ICD-10), the treatment procedures they have undergone, and their prescription information for healthcare research. The study was conducted using the HIRA data from January 1, 2011 to December 31, 2015. All personal information was anonymized before analysis. Thus, informed consent was not required. The protocol of the study was exempted form review by the Institutional Review Board of Seoul National University Hospital and the Seoul National University College of Medicine (IRB number: 1611-056-807).

The subjects included patients who were 18 years of age and above, were prescribed at least one antidepressant, and were diagnosed with depression (F32, F33, F34.1) within 30 days from the index date (the day when the first antidepressant was prescribed) in the year 2012. Depression diagnoses did not need to be a primary diagnosis. Subjects were followed-up on until December 31, 2015. The participants with the following four criteria were excluded from the study. 1) Subjects who had already been prescribed antidepressants within 120 days before the index date. This was done to select only patients with new antidepressant prescription. 2) Subjects who had been diagnosed with psychotic disorder, bipolar disorder, and dementia (F20, F21, F22, F23, F25, F28, F29, F30, F31, F00, F01, F02, F03, and F04) from January 1, 2011 to the index date. Patients with manic episodes (F30) and bipolar affective disorder (F31) were excluded if these were encoded as the primary diagnosis, to avoid including patients given a provisional diagnosis for an insurance claim on mood stabilizers in the absence of definitive evidence for bipolar disorder. 3) Subjects who had not been followed-up for a period of 120 days after the end of the prescription. 4) Subjects with the the exclusion diagnosis of psychotic disorder, bipolar disorder, and dementia during the follow-up period. After exclusion, a total of 752,910 subjects were included in the study. The selection flow of the study participants from the HIRA database is shown in S1 Fig.

Subjects were divided into two groups, short-term and long-term antidepressant users, according to whether they were able to maintain the initial antidepressant for 28 days. The cut-off of 28 days was used to account for the minimum duration of antidepressant use needed to validate its effect before concluding that a patient is either partially responsive or unresponsive to a specific intervention[5]. Short-term users include patients who discontinued antidepressants within 28 days and those who changed antidepressants rapidly without sufficient time to assess their effects over 28 days. Thus, the definition of short-term use addressed the duration of antidepressant use as well as the adequacy of the individual antidepressant prescription duration. Among all the subjects, 458,057 (60.84%) were short-term antidepressant users while 294,853 (39.16%) were long-term antidepressant users.

### Pharmaceutically treated depression (PTD)

PTD was defined as a period when an antidepressant was dispensed to a patient for a depressive disorder to act as a proxy for a depressive episode. We considered that the prescription for an antidepressant was made for a depressive disorder only when the diagnostic code for depression was given within 30 days of the initial prescription of the antidepressant. The end

of an episode of PTD was defined when the subject had not been dispensed an antidepressant for 120 days, met the exclusion criteria, or reached the end date of the study. The numbers of prescriptions, psychiatric admissions, and suicide attempts during the PTD period were also investigated.

### Index date antidepressants

Index date antidepressant prescription was classified as either: monotherapy or polypharmacy. Monotherapy was defined as the prescription of only one antidepressant on the index date, and polypharmacy was defined as the prescription of two or more psychotropic drugs including antidepressants, antipsychotics, and mood stabilizers. When two drugs were prescribed on the index date, it was further distinguished as same class polypharmacy (two antidepressants) and multi-class polypharmacy or augmentation (one antidepressant and one mood stabilizer or antipsychotic) to allow a more detailed picture of the prescription status.

### Confounding variables

Sociodemographic variables, such as age on the index date and sex, were included as confounding variables. Clinical variables including psychiatric and medical comorbidities, were investigated. Patients were defined as having a comorbidity when they were diagnosed with psychiatric comorbidities, such as anxiety disorder, substance-related disorder, obsessive-compulsive disorder, and personality disorders, and non-psychiatric comorbidities, such as cardiovascular diseases, diabetes mellitus, chronic obstructive pulmonary diseases, cancer, stroke, and hypothyroidism within 90 days before the index date. The ICD-10 code for each comorbidity is listed in S1 Table.

Factors related to medical utilization such as insurance coverage, type of medical institutions, and type of medical provider, were investigated. The status of insurance coverage was classified into three categories: NHI subscribers, medical assistances, and patriots and veterans. The latter two groups received partial or complete governmental aid for their medical costs. The types of medical institutions visited were grouped according to the index date visit forms into small hospitals and clinics, and general and tertiary care hospitals. A hospital in the latter group had to have more than 100 beds and at least seven medical departments and specialists. We also examined whether the index date prescription was given by a neuropsychiatric specialist or a general practitioner. This was distinguished using the psychiatric interview charge code.

### Relapse and recurrence

The current study regarded relapse and recurrence as the outcome of the antidepressant treatment rather than the remission of depressive symptoms. Relapses and recurrences were defined as the re-initiation of antidepressant therapy in patients with depression during their follow-up period. While relapses and recurrences have different clinical definitions, we did not make distinctions between relapses and recurrences in the study. By definition, an episode of PTD ends when the subject has not dispensed an antidepressant for 120 days. If the same subject had another PTD, it meant that there was an interval of at least 120 days between the two PTDs. In our study, we included the second PTD in the follow-up period until the year 2015 as a relapse and recurrence.

### Statistical analysis

Comparisons between the short-term and long-term antidepressant users were performed using the Wilcoxon's rank sum test for continuous variables and the chi-square test for

categorical variables. The logistic regression model was performed to find the risk factors for short-term antidepressant use, and odds ratio and 95% confidence intervals were calculated.

Survival curves for relapse and recurrence were generated using the Kaplan-Meier method. Cox proportional hazard models were used to find out the determinants of relapses and recurrences. The forward selection method was used to take into account the multicollinearity of variables for the multivariate models which included the duration of the antidepressant use, demographic characteristics, comorbidity, and clinical variables. Hazard ratios and their 95% confidence intervals were calculated. The proportionality assumption was checked by drawing log-log survival plots to assess whether the assumption was valid. All data management and statistical analyses were performed using SAS Enterprise Guide, version 6.1 (SAS Institute, Cary, NC, USA).

## Results

### Demographic and clinical characteristics

Comparison of the baseline characteristics between the short-term and long-term antidepressant user groups are presented in Table 1. Short-term antidepressant users were younger (52.31 versus 54.40, *p < .001*), composed of more females (66.85% versus 62.235%, *p < .001*), and had fewer psychiatric and medical comorbidities than long-term antidepressant users. Medical utilization factors were significantly different between short-term and long-term antidepressant users.

Table 2 presents the PTD characteristics of short-term and long-term antidepressant users in Korea. The duration of PTD was much shorter (mean: 21.68 days versus 207.09 days; median: 8 days versus 115 days, *p < .001*)among short-term antidepressant users than long-term antidepressant users. There were fewer psychiatric hospitalizations (1.51% versus 5.21%, *p < .001*) and suicide attempts (0.72% versus 3.40%, *p < .001*) among short-term antidepressant users than long-term antidepressant users.

### Initial antidepressant use on index date

The initial status of the antidepressant prescription is presented in Table 3. Among all the patients, 91.07% started their treatment with a single antidepressant. Among all the patients, 432,291 (57.42%) were initially prescribed one TCA, and 174,877 (23.23%) were prescribed one SSRI. A TCA was taken as the initial monotherapy drug by a higher proportion of short-term antidepressant users than long-term antidepressant users (64.25% versus 46.80%). The proportion of patients who were prescribed with polypharmacy was lower in the short-term antidepressant user group than in the long-term antidepressant user group (7.30% versus 11.45%). Same class polypharmacy was more common than augmentation or multiclass polypharmacy (7.5% versus 0.85%) among all the patients. Short-term antidepressant users were initially prescribed a smaller number of antidepressants (1.0 versus 1.11, *p < .001*).

### Factors related to short-term antidepressant use

The crude and adjusted odds ratios for short-term antidepressant use are presented in Table 4. Significant variables in the univariate model included age, sex, type of initial antidepressant, psychiatric and medical comorbidities, clinical specialty, status of insurance coverage, type of hospital the patient initially visited, and prescription of polypharmacy on the index date. After the adjustment of significant variables in the univariate model, the results still confirmed that the odds of short-term antidepressant use was higher when a TCA was used as the initial antidepressant (adjusted OR [aOR] = 1.33, 95% CI = 1.29–1.36), and lower when an SSRI

**Table 1. Comparison of baseline characteristics between short-term and long-term antidepressant users.**

| Characteristics | | Short-term user | Long-term user | p-value |
|---|---|---|---|---|
| n(%) | | 458,057 (60.84) | 294,853 (39.16) | |
| Age | | | | |
| | Mean ± SD | 52.31 ± 16.15 | 54.40 ± 15.76 | <0.001* |
| | Median (Q1,Q3) | 53 (41,64) | 55 (44,67) | |
| | 18–29 | 45,741 (10.00) | 22,718 (7.70) | <0.001†† |
| | 30–39 | 59,663 (13.03) | 32,130 (10.90) | |
| | 40–49 | 84,366 (18.42) | 49,749 (16.87) | |
| | 50–59 | 112,739 (24.61) | 74,021 (25.10) | |
| | 60–69 | 78,535 (17.15) | 60,005 (20.35) | |
| | 70–79 | 60,692 (13.25) | 44,770 (15.18) | |
| | 80- | 16,321 (3.56) | 11,460 (3.89) | |
| Sex | | | | <0.001†† |
| | Male(%) | 151,826 (33.15) | 108,403 (36.77) | |
| | Female (%) | 306,231 (66.85) | 186,450 (63.23) | |
| Psychiatric comorbidity | | | | |
| | Anxiety disorders | 98,348 (21.47) | 86,820 (29.45) | <0.001†† |
| | Obsessive-compulsive disorders | 1,779 (0.39) | 2,229 (0.76) | <0.001†† |
| | Personality disorders | 1,315 (0.29) | 1,627 (0.55) | <0.001†† |
| | Substance use disorders | 8,740 (1.91) | 10,289 (3.49) | <0.001†† |
| Medical comorbidity | | | | |
| | Cardiovascular disease | 144,719 (31.59) | 114,268 (38.75) | <0.001†† |
| | Stroke | 30,433 (6.64) | 30,430 (10.32) | <0.001†† |
| | COPD | 71,853 (15.69) | 48,265 (16.37) | <0.001†† |
| | DM | 67,953 (14.84) | 57,181 (19.39) | <0.001†† |
| | Cancer | 38,015 (8.30) | 31,528 (10.69) | <0.001†† |
| | Hypothyroidism | 14,653 (3.20) | 11,630 (3.94) | <0.001†† |
| Types of medical insurance | | | | <0.001†† |
| | National Health Insurance | 434,979 (94.96) | 272,110 (92.29) | |
| | Medical Assistance | 22,584 (4.93) | 21,129 (7.17) | |
| | Patriots & veterans | 494 (0.11) | 1,614 (0.55) | |
| Types of medical institutions | | | | <0.001†† |
| | Small hospitals and clinics | 335,807 (73.31) | 165,743 (56.21) | |
| | General and tertiary care hospitals | 122,250 (26.69) | 129,110 (43.79) | |
| Prescribed by neuropsychiatric specialists | | | | <0.001†† |
| | NP prescription | 125,765 (27.46) | 119,737 (40.61) | |
| | Non-NP prescription | 332,292 (72.76) | 175,116 (59.39) | |

*SD* standard deviation, *Q1* first quartile, *Q3* third quartile, *COPD* chronic obstructive pulmonary disease, *DM* diabetes mellitus, *NP* neuropsychiatrist specialist.

* Wilcoxon rank sum test

†† Chi-square test

(aOR = 0.73, 95% CI = 0.71–0.75) or a serotonin-norepinephrine reuptake inhibitor (aOR = 0.78, 95% CI = 0.75–0.80) was used. The prescription of polypharmacy on the index date significantly decreased the odds of short-term antidepressant use (aOR = 0.86, 95% CI = 0.84–0.88). Psychiatric and medical comorbidities, except chronic obstructive pulmonary disease, on the index date significantly predicted short-term antidepressant use. Those on medical assistance, patriots, and veterans had decreased odds of short-term antidepressant use

**Table 2. PTD characteristics of short-term and long-term antidepressant users.**

| Characteristics | | Short-term user | Long-term user | p-value |
|---|---|---|---|---|
| PTD duration (days) | | | | |
| | Mean ± SD | 21.68 ± 42.22 | 207.90 ± 230.83 | <0.001* |
| | Median (Q1,Q3) | 8 (5, 18) | 115 (53, 267) | |
| Prescription number during PTD duration | | | | |
| | Mean ± SD | 1.65 ± 1.28 | 8.16 ± 9.94 | <0.001* |
| | Median (Q1,Q3) | 1 (1,2) | 5 (2,10) | |
| Psychiatric admission during PTD (%) | | 6,899 (1.51) | 15,351 (5.21) | <0.001†† |
| Suicide attempt during PTD (%) | | 3,313 (0.72) | 10,027 (3.40) | <0.001†† |

*PTD* pharmaceutically-treated depression, *SD* standard deviation, *Q1* first quartile, *Q3* third quartile.

* Wilcoxon rank sum test

†† Chi-square test

(aOR = 0.64, 95% CI = 0.62–0.65). Conversely, an initial visit to a general doctor (aOR = 1.62, 95% CI = 1.60–0.64) and an initial visit to a small hospital or clinic (aOR = 2.28, 95% CI = 2.25–2.30) increased the odds of short-term antidepressant use.

## Predictors of relapse and recurrence of PTD

The mean time to a relapse or recurrence was longer among short-term antidepressant users than among long-term antidepressant users (504.24 ± 325.03 days versus 402.80 ± 276.49 days).

Table 5 presents the potential risk factors for the time to relapse and recurrence using both, crude and adjusted, models through Cox-proportional regression analyses. Short-term antidepressant use was negatively associated with relapses and recurrences in the univariate model. However, after adjustment, the risk of relapses and recurrences mildly increased with short-term antidepressant use (adjusted hazard ratio [aHR] = 1.062, 95% CI = 1.05–1.08). In the adjusted model, the HR for relapses and recurrences increased as the age increased, except

**Table 3. Index date prescription of antidepressants in short-term and long-term antidepressant users.**

| Characteristics | Total | Short-term user | Long-term user |
|---|---|---|---|
| **1 Drug N(%)** | 424,619 (92.70) | 685,704 (91.07) | 261,085 (88.55) |
| TCA | 432,291 (57.42) | 294,300 (64.25) | 137,991 (46.80) |
| SSRI | 174,877 (23.23) | 88,441 (19.31) | 86,436 (29.31) |
| SARI | 37,354 (4.96) | 21,422 (4.68) | 15,932 (5.40) |
| SNRI | 24,790 (3.29) | 12,042 (2.63) | 12,748 (4.32) |
| NaSSA | 8,060 (1.07) | 3,829 (0.83) | 4,231 (1.43) |
| **2 Drugs (Same-Class Polypharmacy)** | 28,820 (6.29) | 56,495 (7.50) | 27,675 (9.39) |
| SSRI & SARI | 19,594 (2.60) | 9,399 (2.05) | 10,195 (3.46) |
| SSRI & TCA | 15,896 (2.11) | 8,503 (1.86) | 7,393 (2.51) |
| **2 Drugs (augmentation/ Multi-Class Polypharmacy)** | 2,865 (0.63) | 6,430 (0.85) | 3,565 (1.21) |
| **3 drugs** | 1,538 (0.34) | 3,734 (0.50) | 2,196 (0.74) |
| **More than 4 drugs** | 215 (0.05) | 547 (0.07) | 332 (0.11) |
| **Total** | 752,910 | 458,057 | 294,853 |

*TCA* tricyclic antidepressants, *SSRI* selective serotonin reuptake inhibitors, *SARI* serotonin antagonist and reuptake inhibitors, *SNRI* serotonin noradrenaline reuptake inhibitors, *NaSSA* noradrenergic and specific serotonergic antidepressants. Proportion under 1% were not depicted here (See S2 Table.)

**Table 4. Logistic regression for short-term antidepressant users.**

| Characteristics | Crude OR | (95% CI) | p-value | Adjusted OR* | (95% CI) | p-value |
|---|---|---|---|---|---|---|
| Age | 0.99 | (0.99–0.99) | <0.001 | 0.99 | (0.99–0.99) | <0.001 |
| Female gender | 1.17 | (1.16–1.18) | <0.001 | 1.14 | (1.13–1.15) | <0.001 |
| Index date antidepressant prescription | | | | | | |
| TCA | 1.98 | (1.97–2.00) | <0.001 | 1.33 | (1.29–1.36) | <0.001 |
| SSRI | 0.54 | (0.53–0.54) | <0.001 | 0.73 | (0.71–0.75) | <0.001 |
| SARI | 0.70 | (0.69–0.72) | <0.001 | 1.03 | (1.01–1.06) | 0.022 |
| SNRI | 0.57 | (0.55–0.58) | <0.001 | 0.78 | (0.75–0.80) | <0.001 |
| Index date polypharmacy | 0.61 | (0.60–0.62) | <0.001 | 0.86 | (0.84–0.88) | <0.001 |
| Psychiatric comorbidity | | | | | | |
| Anxiety disorder | 0.66 | (0.65–0.66) | <0.001 | 0.82 | (0.81–0.83) | <0.001 |
| Obsessive-compulsive disorders | 0.51 | (0.48–0.55) | <0.001 | 0.63 | (0.59–0.79) | <0.001 |
| Personality disorders | 0.52 | (0.48–0.56) | <0.001 | 0.73 | (0.68–0.79) | <0.001 |
| Substance use disorders | 0.54 | (0.52–0.55) | <0.001 | 0.77 | (0.75–0.80) | <0.001 |
| Medical comorbidity | | | | | | |
| Cardiovascular | 0.73 | (0.72–0.74) | <0.001 | 0.86 | (0.85–0.88) | <0.001 |
| Stroke | 0.62 | (0.61–0.63) | <0.001 | 0.82 | (0.81–0.84) | <0.001 |
| COPD | 0.95 | (0.94–0.96) | 0.0138 | 1.06 | (1.04–1.07) | <0.001 |
| DM | 0.72 | (0.72–0.73) | <0.001 | 0.88 | (0.87–0.89) | <0.001 |
| Cancer | 0.76 | (0.74–0.77) | <0.001 | 0.93 | (0.91–0.94) | <0.001 |
| Hypothyroidism | 0.81 | (0.79–0.83) | <0.001 | 0.86 | (0.83–0.88) | <0.001 |
| Initial visit to non-NP | 1.81 | (1.79–1.82) | <0.001 | 1.62 | (1.60–1.64) | <0.001 |
| Medical assistance, patriots and veterans | 0.64 | (0.62–0.65) | <0.001 | 0.67 | (0.65–0.68) | <0.001 |
| Initial visit to small hospitals and clinics | 2.14 | (2.12–2.16) | <0.001 | 2.28 | (2.25–2.30) | <0.001 |

*OR* odds ratio, *TCA* tricyclic antidepressants, *SSRI* selective serotonin reuptake inhibitors, *SARI* serotonin antagonist and reuptake inhibitors, *SNRI* serotonin noradrenaline reuptake inhibitors, *COPD* chronic obstructive pulmonary disease, *DM* diabetes mellitus, *NP* neuropsychiatrist specialist

*adjusted model included all significant variables in the crude model

among patients who were over 80 years old. The female sex and psychiatric comorbidities consistently increased the aHR for relapses and recurrences. The presence of at least one medical comorbidity had an aHR of 1.06 (95% CI = 1.048–1.075). Patients who were treated by non-psychiatrists had a lower risk of relapses and recurrences than patients who were initially treated by psychiatrists (aHR = 0.69, 95% CI = 0.686–0.699). The aHR of relapses and recurrences was 1.29 (95% CI = 1.271–1.309) among those on medical assistance, patriots, and veterans. An initial visit to small hospitals and clinics rather than to general hospitals had an aHR of 1.29 (95% CI = 1.277–1.309). The prescription of polypharmacy on the initial date marginally increased the risk of relapses and recurrences (aHR = 1.06, 95% CI = 1.041–1.069).

Various characteristics of previous PTD were related to relapses and recurrences. Specifically, psychiatric admission during PTD had a protective effect against relapses and recurrences (aHR = 0.81, 95% CI = 0.786–0.825). Whereas, suicide attempt during the PTD period had an increased risk for relapses and recurrences (aHR = 1.152, 95% CI = 1.121–1.184). A longer duration of previous PTD, especially a duration over 95 days, also predicted a higher risk of relapse (aHR = 2.26; 95% CI = 2.227–2.297).

## Discussion

The main objective of this study was to find the characteristics and clinical outcomes among short-term antidepressant users who quit or changed their medication before the maximum

**Table 5. Cox proportional hazard ratio for relapse and recurrence.**

| Characteristics | Crude HR | (95% CI) | *p*-value | Adjusted HR* | (95% CI) | *p*-value |
|---|---|---|---|---|---|---|
| Age group (reference) 18–29 | | | | | | |
| 30–39 | 1.161 | (1.139–1.184) | <0.001 | 1.188 | (1.165–1.212) | <0.001 |
| 40–49 | 1.376 | (1.351–1.401) | <0.001 | 1.439 | (1.413–1.466) | <0.001 |
| 50–59 | 1.611 | (1.583–1.639) | <0.001 | 1.694 | (1.665–1.725) | <0.001 |
| 60–69 | 1.881 | (1.848–1.914) | <0.001 | 1.96 | (1.924–1.996) | <0.001 |
| 70–79 | 2.015 | (1.978–2.051) | <0.001 | 2.07 | (2.031–2.109) | <0.001 |
| ≥ 80 | 1.671 | (1.627–1.715) | <0.001 | 1.73 | (1.684–1.777) | <0.001 |
| Female gender | 1.234 | (1.224–1.245) | <0.001 | 1.212 | (1.202–1.223) | <0.001 |
| Comorbidities | | | | | | |
| Anxiety disorder | 1.399 | (1.387–1.411) | <0.001 | 1.227 | (1.217–1.238) | <0.001 |
| Obsessive-compulsive disorders | 1.318 | (1.255–1.384) | <0.001 | 1.225 | (1.166–1.287) | <0.001 |
| Personality disorders | 0.988 | (0.927–1.054) | 0.7217 | | | |
| Substance use disorders | 1.328 | (1.299–1.359) | <0.001 | 1.195 | (1.166–1.225) | <0.001 |
| At least one medical comorbidity | 1.155 | (1.141–1.169) | <0.001 | 1.139 | (1.124–1.154) | <0.001 |
| Short-term antidepressant use | 0.629 | (0.624–0.633) | <0.001 | 1.062 | (1.048–1.075) | <0.001 |
| Initial visit to non-NP | 0.698 | (0.693–0.704) | <0.001 | 0.693 | (0.686–0.699) | <0.001 |
| Medical assistance, patriots and veterans | 1.429 | (1.408–1.450) | <0.001 | 1.29 | (1.271–1.309) | <0.001 |
| Initial visit to small hospitals and clinics | 1.235 | (1.224–1.245) | <0.001 | 1.288 | (1.277–1.309) | <0.001 |
| Index date polypharmacy | 1.26 | (1.244–1.277) | <0.001 | 1.055 | (1.041–1.069) | <0.001 |
| PTD characteristics | | | | | | |
| Psychiatric admission during PTD | 1.192 | (1.165–1.219) | <0.001 | 0.805 | (0.786–0.825) | <0.001 |
| Suicide attempt during PTD | 1.548 | (1.507–1.591) | <0.001 | 1.152 | (1.121–1.184) | <0.001 |
| PTD duration(reference: 0–7 days) | | | | | | |
| 8–24 days | 1.212 | (1.197–1.226) | <0.001 | 1.185 | (1.171–1.200) | <0.001 |
| 25–95 days | 1.589 | (1.571–1.607) | <0.001 | 1.581 | (1.559–1.604) | <0.001 |
| 95 days | 2.376 | (2.350–2.401) | <0.001 | 2.262 | (2.227–2.297) | <0.001 |

*HR* hazard ratio, *PTD* pharmaceutically-treated depression, *NP* neuropsychiatrist specialist.

*adjusted model included all significant variables in the crude model

effect of the medication could emerge. The results suggested that short-term antidepressant users occupied a large proportion of patients (60.84%) with depressive disorders who were taking antidepressants in Korea. The results are consistent with previous studies showing that approximately four of ten patients discontinue antidepressants within 30 days [9]. They usually visited small hospitals and clinics initially, and often went to non-psychiatrists rather than psychiatry specialists. They visited hospitals only once or twice per PTD and were often initially prescribed with a single TCA. Their time-to-relapse was longer among compared to adequate users. The type of initial antidepressant, polypharmacy, psychiatric and medical comorbidities, type of insurance coverage, and type of medical institution visited all increased the odds of short-term antidepressant use. Simultaneously, short-term use itself marginally increased the risk of relapse and recurrence of depressive episodes.

To our knowledge, the present study is the first study to focus on the group of patients who were not able to complete 28 days of a specific antidepressant. The guidelines available require a duration of 28 days of using a medication before it can be concluded whether the medication is effective[5]. However, data from the current study implies that the majority of people were not following the guidelines. Considering its retrospective and administrative nature, and the

limited information available, a few careful inferences can be made on why patients quit or switched antidepressants so fast.

Firstly, the majority of short-term antidepressant users probably had milder cases of illness. This can be inferred based on the following findings. The proportion of patients with psychiatric and medical comorbidities on the index date were lower in the short-term antidepressant user group. Since a higher number of comorbidities suggests poorer clinical outcomes in the acute phase of treatment for depressive disorders [22, 23], the short-term antidepressant users can be postulated to have had milder cases of illness. Further, more than 70% of short-term antidepressant users received prescriptions from non-psychiatrists initially even though the Korean medical system gives patients the freedom to choose psychiatric specialists for their initial visit. This also implies that their symptoms were milder since a previous study using administrative data reported that being treated by a psychiatrist reflects more severe depression [24]. Moreover, short-term antidepressant users had shorter PTD durations, fewer psychiatric admissions and suicide attempts, which implies that their symptoms were milder.

Secondly, the initial medication may have affected the discontinuation of antidepressants within a shorter term. Surprisingly, the results of the present study suggested that the initial choice of antidepressants in Korea is TCA, which is in contrast to the results of the study by Bae et al [25] which presents SSRIs as the most commonly prescribed initial antidepressant in Korea. The findings of the current study are consistent with that of the previous study showing that patients whose initial therapy is with an SSRI are more likely to complete treatment over an adequate duration than patients whose initial therapy is with a TCA [26]. This phenomenon may be explained by the underlying pharmacological differences between the two classes of drugs. Previous studies have shown that the efficacy of SSRIs does not differ significantly from that of TCAs. However, TCAs are less tolerable than SSRIs, resulting in TCA users having a higher rate of withdrawal from treatment [27–29]. Moreover, the prescription of TCAs in Korea is strongly related to the status of insurance coverage, the clinical specialty, and region [30]. Considering the status of prescriptions in Korea, the current study revealed that prescriptions of TCAs were related to short-term antidepressant use even after the status of insurance and clinical specialty were adjusted for. The tolerability of TCAs may have caused short-term inadequate use. However, careful interpretation is still needed.

Thirdly, since a high proportion of antidepressants were prescribed by non-psychiatric specialists, it can be presumed that they were used off-label. Antidepressants are effective in the treatment of functional disorders, somatic symptoms, and pain [31–33]. However, in Korea, the HIRA does not approve off-label use of antidepressants. It is possible that doctors give patients the diagnosis of depression in order to use antidepressants for other purposes, even when the patient does not have depressive symptoms, to avoid cutback on health insurance claims. In the Korean insurance system, claims for most antidepressants prescribed by non-psychiatrists for 60 consecutive days are cutback. TCAs, however, can be prescribed for more than 60 days without the insurance claim being cutback even when they are prescribed by non-psychiatrists This, in turn, may influence the preference of the prescription of this class of antidepressants. This tendency would be greater in small hospitals and clinics where the cutbacks on claims may have a greater influence on the salaries of the doctors.

Fourthly, the status of insurance coverage highly affected the duration of antidepressant prescription. Those on medical assistance with low socioeconomic status have a higher incidence and persistence of depression [34–36]. Veterans are vulnerable to psychiatric disorders and have a high prevalence of PTSD and depressive disorders [37, 38]. Hence, they visit hospitals more frequently with more severe symptoms [39]. The presence of more severe symptoms among patients in vulnerable groups may have decreased their odds of using antidepressants

short-term. Moreover, with aid for their medical costs, they were able to visit hospitals when their symptoms remained.

Most of the variables related to relapses and recurrences in the present study were consistent with previous studies [40]. The risk of relapses and recurrences were strongly predicted by old age with many psychiatric and medical comorbidities, which corresponds to the risks seen in previous studies [41, 42]. However, cautious interpretation is needed since a large number of comorbid conditions may lead to care-seeking behavior [18]. An initial visit to a non-psychiatrist not only predicted short-term antidepressant use, but also decreased the risk of relapses and recurrences, which supports the assumption that patients with milder illnesses initially choose to visit non-psychiatrists rather than psychiatric specialists. An initial visit to a small hospital was associated with short-term antidepressant use, while, it increased the risk ratio for future relapses and recurrences. This phenomenon should be the target of further investigation since continuing antidepressants for a sufficient period of time among these patients may help prevent relapses and recurrences among them. Those on medical assistance, patriots, and veterans have increased risks of relapses and recurrences, suggesting that their symptoms are more severe, while their thresholds to visit hospitals are lower. Psychiatric admissions during the previous PTD had a protective effect on the rate of relapses and recurrences which implies that even though patients with more severe illness are admitted into the psychiatric ward, their experiences of admission and intensive care may have helped in preventing relapses and recurrences. In contrast, suicide attempts during a PTD, and the duration of the PTD, which are proxies for the severity of the symptoms, increased the risk of relapses and recurrences.

## Limitations

There were several limitations to the current study. Since the study was based on claimed data, the design had innate limitations similar to other claim-based studies. The study was based on various operational definitions such as "PTD" and "short-term antidepressant use". Moreover, identifying clinical events from the claimed data was limited, especially the response to a specific antidepressant. Relapses and recurrences were the only outcomes that could be measured. Symptom severity and phenomenology of symptoms, which were not included in the claim data, could not be measured. A large number of patients with depression do not return even when their symptoms recur[43]. This implies that the rate of relapses and recurrences in the claimed data would have been underestimated. In addition, most patients do not take the antidepressants as prescribed [44], which the claim data could not have detected. The doctor-patient relationship, which is one of the most important things in adherence to antidepressants, could not be assessed in this study.

## Conclusions

To our knowledge, it was the first study to focus on people who did not take antidepressants properly for even 28 days. The study covered almost the entire Korean population through the HIRA database. Hence, the results may be applicable to all adult patients with depressive disorder. The results suggested that short-term antidepressant use was related to multiple sociodemographic and medication utilization factors. It seems that short-term antidepressant users included milder cases of depressive disorder and there was a minimally increased risk of relapses and recurrences after several factors were adjusted for. By elucidating short-term antidepressant users, more individualized plans can be created, and thus, the direct and indirect burdens of depression can be reduced.

## Supporting information

**S1 Fig. Inclusion flow.**
(PDF)

**S1 Table. The ICD-10 codes of psychiatric and non-psychiatric comorbidities.**
(PDF)

**S2 Table. Index date prescription of antidepressants of inadequate and adequate antidepressant users in Korea.**
(PDF)

## Acknowledgments

The authors would like to thank the mental health professionals and subjects who were involved in the project.

## Author Contributions

**Conceptualization:** Namwoo Kim, Daun Shin, C. Hyung Keun Park, Hyeyoung Kim, Sung Joon Cho, Jae Won Lee, Eun Young Kim, Yong Min Ahn.

**Data curation:** Namwoo Kim, Boram Yang, Yong Min Ahn.

**Formal analysis:** Boram Yang.

**Funding acquisition:** Namwoo Kim, C. Hyung Keun Park, Sung Joon Cho, Jae Won Lee, Yong Min Ahn.

**Investigation:** Sung Joon Cho, Yong Min Ahn.

**Methodology:** Min Ji Kim, Daun Shin, Sang Jin Rhee, C. Hyung Keun Park, Hyeyoung Kim, Boram Yang.

**Project administration:** Sung Joon Cho, Yong Min Ahn.

**Supervision:** Sang Jin Rhee, C. Hyung Keun Park, Yong Min Ahn.

**Validation:** Min Ji Kim.

**Visualization:** Min Ji Kim.

**Writing – original draft:** Min Ji Kim.

**Writing – review & editing:** Daun Shin, Sang Jin Rhee, C. Hyung Keun Park, Hyeyoung Kim, Sung Joon Cho, Jae Won Lee, Eun Young Kim, Boram Yang, Yong Min Ahn.

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
