## [Decision Letter · Decision Letter 0]

24 Jul 2019

PONE-D-19-14920

The epidemiology of antidepressant use in South Korea: Does short-term antidepressant use affect the relapse and recurrence of depressive episodes?

PLOS ONE

Dear Dr. Ahn,

Thank you for submitting your manuscript to PLOS ONE. After careful consideration, we feel that it has merit but does not fully meet PLOS ONE’s publication criteria as it currently stands. Therefore, we invite you to submit a revised version of the manuscript that addresses the points raised during the review process.

Please pay particular attention to grammatical formatting throughout the manuscript. There are several places throughout where proofreading is needed to address grammar for flow and clarity. In addition, please make sure that language describing the study is in past tense (e.g., last paragraph of the introduction) as the study has already occurred. And when acronyms are introduced for the first time in the text (e.g., TCA), they should be spelled out for the reader.

As noted by the reviewer's comments below, there are additional clarifications needed in the presentation of information in the method and results sections. In addition, I will add that the language regarding the exclusion criteria needs to be revised as the text states that there are "three criteria", but four criteria are listed. And "claimed data" should be revised as "claims data" as the data are based on medical billing claims (i.e., ICD codes). 

We would appreciate receiving your revised manuscript by Sep 07 2019 11:59PM. To enhance the reproducibility of your results, we recommend that if applicable you deposit your laboratory protocols in protocols.io, where a protocol can be assigned its own identifier (DOI) such that it can be cited independently in the future. For instructions see: http://journals.plos.org/plosone/s/submission-guidelines#loc-laboratory-protocols

We look forward to receiving your revised manuscript.

Kind regards,

Sarah A. Arias

Academic Editor

PLOS ONE

Journal Requirements:

Reviewers' comments:

Reviewer's Responses to Questions

**Comments to the Author**

1. Is the manuscript technically sound, and do the data support the conclusions?

Reviewer #1: Yes

2. Has the statistical analysis been performed appropriately and rigorously? 

Reviewer #1: Yes

3. Have the authors made all data underlying the findings in their manuscript fully available?

Reviewer #1: Yes

4. Is the manuscript presented in an intelligible fashion and written in standard English?

Reviewer #1: Yes

5. Review Comments to the Author

Reviewer #1: Title: The epidemiology of antidepressant use in South Korea: Does short-term antidepressant use affect the relapse and recurrence of depressive episodes?

Summary of the paper

This study investigated various factors (e.g. age, insurance, types of visited hospitals, PTD, prescriptions) related to short-term antidepressant use by using the population based clinical data. The influence of short-term antidepressant use on relapses and recurrences of depression in the future was also investigated. Results showed that short-term antidepressant users were different in various factors (e.g. types of visited hospitals, prescription, the severity of symptom) compared to long-term antidepressant users. Moreover, it was found that the time to relapse in short-term antidepressant users was shorter than in long-term antidepressant users.

General conceptual or design problems

The manuscript addresses clinically important findings in that larger portion of antidepressant users in South Korea was found to be short-term users. Although the paper includes necessary information of the study overall, there are some points that authors could consider. Points for authors to consider were described with details below.

Specific problems/Areas more or less detailed coverage:

Abstract

Abstract is a good summary of this study by including sufficient overall information.

Introduction

The rationale of study was described well in introduction which included information about social importance of depression and antidepressants treatment in South Korea.

Methods

1. Information of data used in this study and demographic information for participants were well presented in Methods. (Exclusion criteria were also clearly stated.) (Participants=more than 18 years; depression diagnoses die not need to be a primary diagnosis.) Although ‘Subjects’ of ‘Methods’ includes information of the data, used in this study, and exclusion criteria, it was unclear of the final sample of participants. Moreover, it was stated that participants were classified as short-term and long-term antidepressant users in Methods of Abstract. Thus, it would be better for authors to move the number of participants for each group, presented in demographic information in ‘Results’, into ‘Subjects’ of ‘Methods’ and to combine this section with the part which defines short-term and long-term users.

2. As this study includes several variables, it would be better to classify variables into socio-demographic, clinical and medial utilization factors, which were stated to affect the duration of antidepressant use in the section of Methods in Abstract.

Results

1. Information, which was presented in tables, was presented in texts. The duplication of the information (e.g. statistical values) between tables and texts should be avoided.

2. Results include lots of information of analyses. Thus, although some key findings of this study were described in the first paragraph of discussion, it would be better to add additional paragraph for the summary of key findings.

Discussion

Findings of studies were interpreted by being supported by relevant literatures and the clinical significance of this study was also stated.

6. PLOS authors have the option to publish the peer review history of their article (what does this mean?). If published, this will include your full peer review and any attached files.

Reviewer #1: No

---

## [Author Response · Author response to Decision Letter 0]

12 Aug 2019

Dear reviewer of the manuscript “The epidemiology of antidepressant use in South Korea: Does short-term antidepressant use affect the relapse and recurrence of depressive episodes?”. 

Thank you for your detailed review and comments. I have provided answers to the comments below. 

Thank you. 

#1. Please pay particular attention to grammatical formatting throughout the manuscript. There are several places throughout where proofreading is needed to address grammar for flow and clarity. In addition, please make sure that language describing the study is in past tense (e.g., last paragraph of the introduction) as the study has already occurred. And when acronyms are introduced for the first time in the text (e.g., TCA), they should be spelled out for the reader.

Answer: I again double checked with the grammar and checked the points you have raised. Thank you.

#2. There are additional clarifications needed in the presentation of information in the method and results sections. In addition, I will add that the language regarding the exclusion criteria needs to be revised as the text states that there are "three criteria", but four criteria are listed. And "claimed data" should be revised as "claims data" as the data are based on medical billing claims (i.e., ICD codes).

Answer: Thank you for your comments. I changed the criteria “tree” into “four”, and change ‘claimed data’ to ‘claims data’.

#3. Information of data used in this study and demographic information for participants were well presented in Methods. (Exclusion criteria were also clearly stated.) (Participants=more than 18 years; depression diagnoses die not need to be a primary diagnosis.) Although ‘Subjects’ of ‘Methods’ includes information of the data, used in this study, and exclusion criteria, it was unclear of the final sample of participants. Moreover, it was stated that participants were classified as short-term and long-term antidepressant users in Methods of Abstract. Thus, it would be better for authors to move the number of participants for each group, presented in demographic information in ‘Results’, into ‘Subjects’ of ‘Methods’ and to combine this section with the part which defines short-term and long-term users.

Answer: Thank you for your comments. I agree that it would be more clear to present the final participants of each group in the ‘Subject’ part of the method. I also combined this section with the ‘Short-term and long-term antidepressant use’ part for further clarifications. I clearly mentioned the final sample of participants. I added the following part in the last paragraph of the ‘subject’ part:

“Subjects were divided into two groups, short-term and long-term antidepressant users, according to whether they were able to maintain the initial antidepressant for 28 days. The cutoff of 28 days was used to account for the minimum duration of antidepressant use needed to validate its effect before concluding that a patient is either partially responsive or unresponsive to a specific intervention[5]. Short-term users include patients who discontinued antidepressants within 28 days and those who changed antidepressants rapidly without sufficient time to assess their effects over 28 days. Thus, the definition of short-term use addressed the duration of antidepressant use as well as the adequacy of the individual antidepressant prescription duration. Among all the subjects, 458,057 (60.84%) were short-term antidepressant users while 294,853 (39.16%) were long-term antidepressant users.”

#4. As this study includes several variables, it would be better to classify variables into socio-demographic, clinical and medial utilization factors, which were stated to affect the duration of antidepressant use in the section of Methods in Abstract.

Answer: Thank you for your comments. I grouped variables in the ‘Confounding variables’ part as you recommended for further clarification. 

#5. Information, which was presented in tables, was presented in texts. The duplication of the information (e.g. statistical values) between tables and texts should be avoided.

Answer: Thank you for your comments. I erased some duplications and only left comparatively important information in texts. 

#6. Results include lots of information of analyses. Thus, although some key findings of this study were described in the first paragraph of discussion, it would be better to add additional paragraph for the summary of key findings.

Answer: Thank you for your comments. I revised the first paragraph of the discussion part to add key findings in the article. The following sentences were added to the paragraph: “The type of initial antidepressant, polypharmacy, psychiatric and medical comorbidities, type of insurance coverage, and type of medical institution visited all increased the odds of short-term antidepressant use. Simultaneously, short-term use itself marginally increased the risk of relapse and recurrence of depressive episodes.”

---

## [Editor Report · Decision Letter 1]

9 Sep 2019

The epidemiology of antidepressant use in South Korea: Does short-term antidepressant use affect the relapse and recurrence of depressive episodes?

PONE-D-19-14920R1

Dear Dr. Ahn,

We are pleased to inform you that your manuscript has been judged scientifically suitable for publication and will be formally accepted for publication once it complies with all outstanding technical requirements.

With kind regards,

Sarah A. Arias

Academic Editor

PLOS ONE

Additional Editor Comments (optional):

- Line 92 should read "...study was exempted **from** review..."

- Line 98 should read "Subjects were **followed-up on until **December..."

- Line 99 should read "...already been **prescribed antidepressants**..."

- Line 106, If the "above exclusion diagnosis" for criteria #4 is patients with manic episodes, this information should be moved to this section so it is clear to what "above exclusion diagnosis" you are referring.

---

## [Editor Report · Acceptance letter]

17 Sep 2019

PONE-D-19-14920R1 

The epidemiology of antidepressant use in South Korea: Does short-term antidepressant use affect the relapse and recurrence of depressive episodes? 

Dear Dr. Ahn:

I am pleased to inform you that your manuscript has been deemed suitable for publication in PLOS ONE. Congratulations! Your manuscript is now with our production department. 

With kind regards,

on behalf of

Dr. Sarah A. Arias 

Academic Editor

PLOS ONE